Comparison and correlation of cervical proprioception and muscle endurance in general joint hypermobility participants with and without non-specific neck pain—a cross-sectional study

http://orcid.org/0000-0001-6638-0585 Reddy Ravi Shankar rshankar@kku.edu.sa
Tedla Jaya Shanker
Alshahrani Mastour Saeed
Asiri Faisal
Kakaraparthi Venkata Nagaraj
Medical Rehabilitation Sciences, King Khalid University , Abha, Aseer , Saudi Arabia
Mendez-Rebolledo Guillermo
Electronic publication date: 2022 Mar 11
Publication date: 2022
Volume: 10
Electronic Location ID: e13097
Received 2021 Nov 22; Accepted 2022 Feb 20
Copyright: © 2022 Reddy et al.
Copyright year: 2022
Copyright holder: Reddy et al.
License: This is an open access article distributed under the terms of the Creative Commons Attribution License, which permits unrestricted use, distribution, reproduction and adaptation in any medium and for any purpose provided that it is properly attributed. For attribution, the original author(s), title, publication source (PeerJ) and either DOI or URL of the article must be cited.
License URL: https://creativecommons.org/licenses/by/4.0/

Keywords: Hypermobility syndrome, Joint kinaesthesia, Joint position sense, Cervical muscle endurance, Sensorimotor deficits

Funding: King Khalid University RGP. 2/40/42 This study was funded by King Khalid University (Grant No: RGP. 2/40/42). The funders had no role in study design, data collection and analysis, decision to publish, or preparation of the manuscript.

==============================
Background

Cervical proprioception and muscle endurance are essential for maintaining cervical functional joint stability. Proprioception and muscle endurance may be impaired in those with general joint hypermobility (GJH). Examining these aspects is crucial. This study’s aims are to (1) compare the cervical joint position error (JPE) and muscle endurance holding capacities in GJH individuals with and without non-specific neck pain (NSNP) (2) to assess the relationship between hypermobility Beighton scores, cervical JPE’s, and muscle endurance in GJH individuals with and without NSNP.

Methods

In this cross-sectional comparative study, 33 GJH participants with NSNP (mean age 21.7 ± 1.8 years) and 35 asymptomatic participants GJH (mean age 22.42 ± 1.7 years) participated. Beighton’s score of ≥4 of 9 tests was used as criteria to diagnose GJH. Cervical JPEs were estimated in degrees using a cervical range of motion device, and muscle endurance (flexor and extensor) were estimated in seconds using a stopwatch.

Results

GJH participants with NSNP showed significantly larger cervical JPEs (p < 0.001) and decreased muscle endurance holding times (p < 0.001) compared to asymptomatic participants. Beighton hypermobility scores showed a significant moderate positive correlation with cervical JPEs (flexion: r = 0.43, p = 0.013), left rotation: r = 0.47, p = 0.005, right rotation: r = 0.57, p = 0.001) in NSNP individuals. Also, Beighton hypermobility scores showed a moderate negative correlation with muscle endurance in NSNP (flexor muscles: r = −0.40, p = 0.020, extensor muscles: r = −0.41, p = 0.020, and asymptomatic individuals (flexor muscles: −0.34, p = 0.045, extensor muscles: r = −0.45, p = 0.007).

Conclusion

GJH individuals with NSNP showed increased cervical JPEs and reduced muscle endurance compared to asymptomatic. Individuals with GJH with higher Beighton scores demonstrated increased cervical JPEs and reduced neck muscle endurance holding ability. In clinical practice, therapists should be aware of these findings, incorporate proprioceptive and muscle endurance assessments, and formulate rehabilitation strategies for NSNP individuals with GJM.

Introduction

General joint hypermobility (GJH) is an inherited condition that may predispose to musculoskeletal pain (Alsiri, 2017). Mutations in the genes that code for collagen, elastin, and tenascin are associated with this condition (Alsiri, 2017). The prevalence of hypermobility syndrome in adults ranges from 0.6% to 31.0% (Demes, McNair & Taylor, 2020). The hypermobility in females is 1.5 to 3 times more when compared to males (Demes, McNair & Taylor, 2020). GJH allows the joints to move beyond their normal range of motion, leading to increased laxity, joint instability, and severe injuries (Demes, McNair & Taylor, 2020). Skin, tendon, bone, ligament, and cartilage have a considerable amount of tensile strength and are more likely to fail mechanically in hypermobile subjects compared to others. This unavoidably has a detrimental effect on many individuals who are drawn to performing physically demanding pursuits where physical demands may surpass the body’s capacity to bear them (Molander et al., 2020).

Together with the visual and vestibular systems, the proprioceptive system is crucial in maintaining balance and joint stability (Alahmari et al., 2017a; Riemann & Lephart, 2002; Uzunkulaoğlu & Çetin, 2019). Previous studies have shown impaired proprioception in the peripheral joints in individuals with GJH (Smith et al., 2013; Uzunkulaoğlu & Çetin, 2019). Decreased proprioceptive acuity might lead the hypermobile joints to instability and increased risk of injury (Smith et al., 2013). In addition, previous research demonstrates that impaired proprioception is crucial in developing and maintaining pain, tissue injury, and degenerative joint diseases (Alahmari et al., 2020a; Alahmari et al., 2017b; Alahmari et al., 2017c; Asiri et al., 2021; Kristjansson & Treleaven, 2009; Reddy et al., 2021a; Reddy et al., 2020). Therefore, assessing cervical proprioception is vital in evaluating and managing subjects with GJH.

In both static and dynamic circumstances, neck muscles play a vital function in supporting the cervical spine (Fountain, Minear & Allison, 1966; Grimmer, 1994; Schieppati, Nardone & Schmid, 2003; Winters & Peles, 1990). The deep cervical and dorsal neck muscles create a sleeve that protects the cervical spine from gravitational forces to maintain stable neck postures during activities of daily living. Impaired cervical muscle function, it is thought, will disrupt the balance between the anterior and posterior aspects of the neck, resulting in a loss of cervical lordosis and consequently tend to contribute to cervical dysfunction. Pain and fatigue are common complaints in individuals with GJH (Reddy, Maiya & Rao, 2012). Neck muscle fatigue may contribute to altered motor control and reduced proprioceptive sensibility at the neck in individuals with GJH. Different authors demonstrated that cervical reposition sense improved with neck muscular endurance retraining (Jull et al., 2007; Rezasoltani et al., 2010). This finding shows that proprioception and muscle endurance are interrelated and can influence one another. However, the magnitude of these effects may be more influenced by pain in GJH individuals. To date, no researcher had looked at alterations in cervical JPS or muscle endurance capabilities in GJH patients with non-specific neck pain (NSNP) and their associations with hypermobility or vice versa. Rehabilitation therapists’ understanding of these aspects will help assess and manage GJH patients with NSNP. Therefore, the objective of this study is to (1) assess and compare the cervical JPS and muscle endurance capabilities in GJH participants with and without NSNP, (2) to evaluate the relationship between GJH Beighton score and cervical JPS scores, neck muscle endurance capabilities in participants with and without NSNP.

Materials and Methods

Design

This cross-sectional comparative study data was collected between April 2021 to October 2021 at the physical therapy outpatient clinics department, King Khalid University. Research Ethics Committee at King Khalid University (HAPO-06-B-001) reviewed and accepted this study protocol (ECM#2021-4404).

Participants

Thirty-three GJH individuals presenting with NSNP were referred to the physical therapy clinic by an orthopedic doctor. Participants have included if the Beighton score was ≥4/9, with NSNP aged between 20 and 45 and able to follow physical therapists commands. NSNP participants were excluded (1) if they had a history of previous surgery, (2) had signs of cervical myelopathy, (3) had general fibromyalgia symptoms, (4) neurological disease, and (5) Ehlers-Danlos Syndrome. This study recruited 35 GJH asymptomatic participants aged between 20 and 30 years who volunteered to participate and followed the physical therapist’s instructions. This research followed the principles of the Helsinki Declaration. All participants signed informed consent before the commencement of the study and after being briefed about the study methods.

Outcome measures

Beighton score

The Beighton score is a prominent hypermobility screening tool (Malek, Reinhold & Pearce, 2021). This nine-point scale needs five maneuvers, four passive bilateral and one active unilateral. It was created for epidemiological research to detect hypermobility in populations (Malek, Reinhold & Pearce, 2021). Across all cultures and ages, the Beighton score has been utilized to characterize generalized joint laxity (Malek, Reinhold & Pearce, 2021). Several prevalence studies use various cutoffs, ranging from three to six hypermobile joints (thumbs, little fingers, elbows, knees, and the trunk), and others only analyze the dominant side. However, a score >4 out of 9 indicates the presence of hypermobility (Malek, Reinhold & Pearce, 2021). The Beighton scale’s components included in this study are (1) Passive—dorsiflexion and hyperextension of the fifth metacarpophalangeal joint in a passive manner beyond 90° (Bilateral), (2) Passive—apposition of the thumb to the flexor aspect of the forearm (bilateral), (3) Passive—hyperextension of the elbow beyond 10° (bilateral), (4) Passive—hyperextension of the knee beyond 10° and (5) Active—forward flexion of the trunk with the knees fully extended so that the palms of the hands rest flat on the floor.

Cervical proprioception (reposition accuracy) measurement

Cervical proprioceptive accuracy is estimated as joint position error (JPE) in degrees. The cervical target position sense is measured using a cervical range of motion (CROM) device. The full cervical range of motion in each direction is recorded, and 50% of the available range was selected as the target position the individuals had to reposition.

Individuals sat straight on a stool and both feet flat on the ground. The examiner secured the CROM device on the participant’s head and asked them to determine their self-selected neutral head position (Fig. 1). Following that, the examiner calibrated the CROM device to zero position. All the participants closed their eyes before the commencement of the test. Next, the physical therapist moved the participant’s head into the target position (50% of the maximum ROM) and held it for 5 s before asking them to memorize this target position. Following this, the examiner guided the participant’s head back to the neutral (beginning) position. Following this, the individuals were instructed to move their heads to the previously memorized target position. After the individuals positioned their heads to the target position, the precision of their repositioning was measured in degrees (JPE). The sense of target head repositioning was evaluated in the cervical flexion, extension, and left and right rotation directions. The test was repeated thrice in each direction, and the mean of these trials was used for analysis. No additional feedback was provided to the participants throughout the testing period. The order of testing directions was randomized using a simple chit method.

Figure 1 Evaluation of cervical joint position errors using a cervical range of motion device.

Cervical flexor endurance testing

The test was performed with the individuals lying in supine and crook lying positions (Cagnie et al., 2007; Harris et al., 2005). A JTech Dualer IQ Digital Inclinometer (JTech Medical, Salt Lake City, UT, USA) was placed on the lateral aspect participant’s forehead and secured with a Velcro (Fig. 2). When compared to an isokinetic dynamometer, the digital inclinometer had a high level of validity (ICC = 1.0, SEM = 0.09, p = 0.001), and there was excellent intra- and inter-tester reliability for reading the inclinometer (ICC = 1.0, SEM = 0.85, p = 0.001) (Romero-Franco et al., 2019; Romero-Franco, Montaño-Munuera & Jiménez-Reyes, 2017). The participants raised their head and neck until the head was roughly 2.5 cm off the plinth while keeping the chin retracted and held isometrically to the chest, as shown in Fig. 1. The participants were instructed to maintain this position as long as they possibly could. The test was terminated if (1) the participant’s head touched with the investigator’s palm for more than 1 s, (2) the skin folds began to separate due to a loss of chin tuck, (3) could not maintain the head in the horizontal position (>5° variation as measured by digital inclinometer), (4) the participant showed a desire to end the test due to exhaustion or pain. The muscle endurance holding time was recorded in seconds using a stopwatch, and the reason for terminating the test was noted in a logbook. Each endurance test was repeated three times, and an average of three values was used for analysis. Between measurements, a minimum of 5 min of resting time was allowed.

Figure 2 Measurement procedure of cervical flexor endurance.

Cervical extensor endurance testing

The endurance test was adapted from Lee, Nicholson & Adams (2005). The participants were asked to lie prone on the examination couch with their heads protruding from the examination couch and their heads supported by the examiner. A strap was used and wrapped around the participant’s thoracic spine at the level of T7 to stabilize the thoracic spine (Fig. 3). A digital inclinometer was strapped to the participant’s head to measure the participant’s head’s alignment in the horizontal plane. A 2 kg weight was placed on the participant’s head and secured around the participant’s forehead using tape, which the examiner initially supported. Extensor endurance was measured by the examiner slowly leaving the patient’s head along with 2 kg weight, allowing the weight to hang just above the floor as a pendulum freely. The participants were asked to maintain this head position with the chin steadily retracted as long as possible. The test terminated if (1) the participant’s head was tilted, or its position shifted more than 5 degrees away from the horizontal plane (as assessed by a digital inclinometer), (2) the participant was unable to maintain the test position due to exhaustion or pain during the test. Each endurance test was repeated three times, and an average value was used for analysis. Between measurements, a minimum of 5 min of resting time was allowed.

Figure 3 Measurement procedure of cervical extensor endurance.

Visual Analog Scale (VAS): The current level of neck pain was measured on a 100 mm continuous scale, with “0” indicating no pain and “100” indicating the most excruciating pain. The participants mark a point on the scale to represent their current pain intensity. The VAS is a reliable tool widely used to assess pain intensity in different cervical disorders (Parazza et al., 2014; Tishelman et al., 2019).

A single examiner who has experience as a musculoskeletal physical therapy specialist for more than 15 years assessed all the outcome measures, and the examiner was blinded to the group allocation.

Statistical analysis

The study data were analyzed using SPSS version 22.0 (IBM Corp., Armonk, NY, USA). The sample Shapiro–Wilk test was used to determine the normal distribution of the data. An independent t-test was used to compare the cervical JPE’s and muscle endurance in participants with GJH with and without NSNP. In addition, we calculated the effect size in terms of Cohen’s d. Minimal detectable change (MDC) is computed to differentiate between random measurement error and real change. MDC was calculated as follows: (Standard Error Mean (SEM) × 1.65 × √2) (Furlan & Sterr, 2018). Pearson’s correlation coefficient (r) was used to assess the association between hypermobility Beighton scores and cervical JPE’s, neck muscle endurance capabilities in GJH individuals with and without NSNP. According to Schober, Boer & Schwarte (2018), we considered this statistic as fair when the correlation (r) value was less than 0.30, moderate when the r value was between 0.31 and 0.60, and good when the r-value is more than 0.60. General Linear Model was used to verify if there are interactions between groups × Beighton scores for each outcome (cervical JPE’s and cervical muscle endurance) and to see interactions between group × Beighton × gender. A 95% confidence level was used to investigate statistical significance, and a p-value of ≤0.05 was considered statistically significant.

Results

Demographics

Thirty-three NSNP (mean age: 21.7 years) and 35 asymptomatic (mean age: 22.4 years) participants were enrolled in this study. Table 1 summarizes the descriptive and demographic characteristics of the study population. Age, height, weight, BMI, and Beighton scores did not differ across groups (all p > 0.05). The Shapiro–Wilk test revealed that the study variables were normally distributed.

Table 1 Demographic characteristics of the study population.

Variables	Neck pain group (n = 33)
(Mean ± SD)	Asymptomatic group (n = 35)
(Mean ± SD)	p-value	
Age (yrs.)	21.7 ± 1.8	22.42 ± 1.7	0.13	
Height (Mts)	1.7 ± 0.1	1.6 ± 0.1	0.74	
Weight (kg)	62.6 ± 12.6	64.6 ± 14.6	0.55	
BMI (kg/m2)	22.8 ± 3.9	23.7 ± 4.1	0.38	
Beighton score (0 to 9)	5.70 ± 1.0	6.09 ± 1.04	0.12	
Pain Intensity (0 to 10 cm)	4.6 ± 1.0	–	–	
NDI score (%)	22.1 ± 3.9	–	–	
Note:

SD, standard deviation; BMI, body mass index; NDI, neck disability index.

The difference in cervical JPEs and cervical muscle endurance

The magnitude of cervical JPEs was significantly larger in the NSNP group (p < 0.001) when compared to the asymptomatic group (Table 2). These differences were seen in all the directions tested. The magnitude of JPE’s was largest in extension direction (NSNP group: 6.45° ± 1.20°, asymptomatic group: 2.29° ± 1.32°) compared to other directions tested. The SEM ranged from 0.29° to 0.39°, and MDC ranged from 0.67° to 0.91° (Table 2). The effect size (Cohen’s d) ranged between 0.46 to 0.73.

Table 2 Comparison of JPE between NSNP and asymptomatic groups.

Variables	NSNP group (n = 33) (Mean ± SD)	Asymptomatic group (n = 35) (Mean ± SD)	Cohen’s d	95% CI of the difference	SEM	MDC	p value	
Lower	Upper	
JPE in flexion (°)	4.94 ± 1.52	1.97 ± 1.71	0.46	2.18	3.75	0.39	0.91	<0.001	
JPE in extension (°)	6.45 ± 1.20	2.29 ± 1.32	0.73	3.56	4.78	0.31	0.72	<0.001	
JPE in left rotation (°)	5.15 ± 1.37	1.14 ± 1.12	0.72	3.40	4.61	0.30	0.70	<0.001	
JPE in right rotation (°)	5.21 ± 1.29	1.97 ± 1.04	0.67	2.67	3.81	0.29	0.67	<0.001	
Cervical flexor endurance (s)	44.39 ± 4.74	60.20 ± 4.37	0.75	−18.01	−13.60	1.10	3.03	<0.001	
Cervical extensor endurance (s)	72.30 ± 13.84	163.63 ± 22.74	0.91	−102.38	−87.70	3.68	10.17	<0.001	
Note:

JPE, Joint position error; NSNP, non-specific neck pain; Cohen’s d, effect size; CI, Confidence Interval; SEM, Standard error of measurement; MDC, Minimal detectable change.

The endurance holding capacities of the cervical flexor and extensor muscles was significantly lower in the NSNP group than in the asymptomatic group (p < 0.001). The NSNP group’s mean cervical flexor endurance holding capacity was 44.39 ± 4.74 s, and the asymptomatic group was 60.20 ± 4.37 s (Table 2). The SEM was 1.10, MDC was 3.03 s, and the Cohen’s d was 0.75 (Table 2). The mean extensor endurance capacity in the NSNP group was 72.30 ± 13.84 s, and the asymptomatic group was 163.63 ± 22.74 s (Table 2). The SEM was 3.68, MDC was 10.17 s, and the Cohen’s d was 0.91 (Table 2).

Relationship between Beighton score and cervical JPEs, cervical muscle endurance

The results of the Pearson’s correlation showed a significant positive moderate correlation between Beighton score and cervical JPEs in the flexion (r = 0.43, p = 0.013), left rotation (r = 0.47, p = 0.005) and right rotation (r = 0.57, p = 0.001) directions in NSNP group (Table 3). The results indicate that the greater is the hypermobility, as indicated by the Beighton score, the greater are the cervical JPEs in the NSNP group. There were no significant correlations between Beighton’s score and cervical JPEs in the asymptomatic group (Table 3).

Table 3 Correlation between Beighton score and JPE’s, neck flexor, and extensor endurance.

	Beighton score	
	NSNP group (n = 33)	Asymptomatic group (n = 35)	
Variables	r	p	r	p	
JPE in Flexion (°)	0.43	0.013	0.08	0.630	
JPE in extension (°)	0.25	0.156	0.153	0.380	
JPE in left Rotation (°)	0.47	0.005	0.26	0.125	
JPE in right rotation (°)	0.57	0.001	0.16	0.357	
Neck flexor endurance (s)	−0.40	0.020	−0.34	0.045	
Neck extensor endurance (s)	−0.41	0.020	−0.45	0.007	
Note:

NSNP, non-specific neck pain; JPE, Joint position error; NDI, Neck disability index.

There was a statistically significant moderate negative correlation observed between Beighton score and cervical muscle endurance holding capacities in both NSNP group (flexor muscles: r = −0.40, p = 0.020, extensor: r = −0.41, p = 0.020) and asymptomatic group (flexor muscles: −0.34, p = 0.045, extensor muscles: r = −0.45, p = 0.007). In both NSNP and asymptomatic groups, the results indicate that individuals with higher Beighton scores had reduced neck flexor and extensor endurance capabilities (Table 3).

There were significant group interactions with Beighton scores for JPEs (p < 0.001) and muscle endurance (p < 0.001). The NSNP group individuals with hypermobility have larger JPEs and lower cervical muscle endurance capabilities. Table 4 shows the pattern of interactions. Also, there were significant gender interactions between groups (p < 0.001). Females had more hypermobility than males in both NSNP and asymptomatic groups.

Table 4 Generalised linear model (GLM) for the interactions of Beighton scores with JPEs and muscle endurance.

Interaction effect with Explanatory variables	B	Standard
error	95% CI
lower, upper	p-value	
Group * JPE in flexion (°)	5.72	0.24	[5.23–6.21]	<0.001	
Group * JPE in extension (°)	5.64	0.32	[4.99–6.30]	<0.001	
Group * JPE in left rotation (°)	6.09	0.23	[5.62–6.56]	<0.001	
Group * JPE in right rotation (°)	5.81	0.33	[5.13–6.49]	<0.001	
Group * Cervical flexor endurance (s)	9.92	1.29	[7.33–12.51]	<0.001	
Group * Cervical extensor endurance (s)	8.45	0.78	[6.89–10.01]	<0.001	
Group * Gender	6.22	0.24	[5.74–6.70]	<0.001	
Note:

B, coefficient; 95% CI, 95% confidence interval; JPE, joint position error.

Discussion

The current study is, to date, the first to assess and compare cervical JPS and neck muscle endurance capabilities in GJH individuals with and without NSNP. Also, this study assessed the correlation between hypermobility scores and proprioceptive JPEs and muscle endurance capabilities in GJH individuals with and without NSNP. The present study results showed that the magnitude of cervical JPE’s was larger, and muscle endurance (flexor and extensor) holding capacities were lower in hypermobile participants with NSNP compared to asymptomatic. The Beighton score showed a moderate positive correlation with cervical JPE’s in the NSNP group and no significant correlations in the asymptomatic group. Also, the Beighton score showed a moderate negative correlation with cervical muscle endurance capacities in NSNP and asymptomatic groups.

Group differences in JPEs and endurance holding capacities

This study’s findings of increased JPEs in NSNP are in accordance with the results of Reddy et al. (2019) study, in which cervical position sense is impaired in the NSNP group compared to the asymptomatic group. The JPEs were significantly larger in flexion, extension, and left and right rotations, comparable to this study’s results. In this study, the magnitude of cervical JPEs in the asymptomatic group was smaller when compared to the Reddy et al. (2019) study. This study showed a range of 1.14° to 2.29° while; the Reddy et al. (2019) study showed 2.36° to 4.48° JPEs. Our study population is hypermobile, and the Reddy et al. (2019) study population is normal; the JPEs were larger in the Reddy et al. (2019) study. It is likely that this is due to the fact that our study participants are younger (mean age: 22.42 years) compared to Reddy et al. (2019) study participants (mean age: 45.07 years). It is well established that with increasing age, the cervical proprioceptive acuity is reduced (Alahmari et al., 2017b, 2017c).

The systematic review results conducted by de Vries et al. (2015) are in accordance with our study, showing that people with neck pain have increased JPEs compared to asymptomatic participants. The increased magnitude of JPE in the neck pain group may be due to the influence of pain that chemically mediates and alters the free nerve ending discharges and produces abnormal afferent information, thus impairing proprioceptive input. In addition, studies have shown a significant association between increased pain intensity and increased cervical proprioception errors (Alahmari et al., 2020b; Reddy et al., 2019).

Our study findings are in accordance with prior research indicating decreased cervical muscle endurance in NSNP patients compared to asymptomatic patients (Alahmari et al., 2019; Amiri Arimi, Ghamkhar & Kahlaee, 2018; Kandakurti et al., 2021; Reddy et al., 2021b). The cervical spine muscles have an abundance of muscle spindle that significantly contributes to afferent motor functionality in maintaining neck endurance (Boyd-Clark, Briggs & Galea, 2002). The reasons for reduced endurance holding capability in NSNP individuals may be increased pain intensity that would reflexively inhibit the muscles and lead to a cycle of pain to weakness (Alshahrani et al., 2022; Reddy et al., 2022; Reddy et al., 2020; Van Wilgen et al., 2003). It is also demonstrated that the type 1 muscle fiber is transformed into type 2 in neck pain individuals, resulting in decreased strength and endurance of neck muscles (Amiri Arimi, Ghamkhar & Kahlaee, 2018; Kandakurti et al., 2021). The mean neck extensor endurance holding capacities in this study (NSNP: 163.63 22 s, asymptomatic: 72.30 13.84 s) is comparable to the findings of Reddy et al. (2021b) (NSNP: 155.88 11.94 s, asymptomatic: 72.00 23.11 s), implying that hypermobile participants with or without NSNP are the comparable population with or without NSNP.

Relationship between hypermobility, proprioception, endurance

This research is the first to evaluate the correlation between hypermobility and cervical JPS, and it demonstrated a moderately positive correlation in the NSNP group. There was no association between hypermobility and cervical JPS in the asymptomatic group. This finding implies that pain may be a factor impacting proprioception in hypermobile persons and that it may be a contributing factor. Ferrell et al. (2004) and Sahin et al. (2008a) showed that hypermobility individuals had larger JPEs in the knee joints. Joint laxity and impaired proprioception may make unstable joints vulnerable to trauma (Sahin et al., 2008a).

Experimentally produced pain models indicated a link between pain and proprioception (Capra & Ro, 2000). Sensitization may modify free nerve ending discharges, resulting in abnormal pain afferents (gamma-motor neuron and muscle spindle), thereby compromising proprioceptive input (Lima et al., 2021). Unlike our investigation, Lee et al. (2008) found no relation between neck pain severity and cervical position sense in neck pain patients; rather, they found a link between pain frequency and JPEs.

This study showed a significant moderate negative correlation between hypermobility and neck muscle endurance capabilities in NSNP and asymptomatic groups. Different studies have demonstrated decreased muscle strength, physical fitness, and impaired proprioception in individuals with hypermobility syndrome (Engelbert et al., 2006; Sahin et al., 2008b; Smith et al., 2013). Decreased muscle strength and endurance will further place the joint for further injury and functional disability (Beighton, Grahame & Bird, 2012). The degree of hypermobility is determined by genetics (Beighton, Grahame & Bird, 2012). Increased ligamentous extensibility is caused by changes in connective tissue such as elastin, collagen tenascin, and fibrillin (Beighton, Grahame & Bird, 2012). Non-inflammatory joint and muscle discomfort is a common symptom of widespread or pauciarticular joint laxity. A previous study showed a correlation between arthralgia and joint hypermobility (Beighton, Grahame & Bird, 2012).

Reduced joint stability and muscle strength, when combined with hypermobility, can play a significant role in the development of neck pain disorders or other musculoskeletal injuries and should be researched further to understand their relationships so that prevention and treatment strategies can be planned in this situation. In addition, hypermobility syndrome has been linked to increased risk of upper and lower limb sports injuries, discomfort, and poor dynamic trunk stability (Jindal et al., 2016; Konopinski, Jones & Johnson, 2012; Simmonds & Keer, 2007). In light of these findings, a complete routine assessment of GJH is required when investigating neck pain issues.

Limitations

This study’s findings should be seen within certain constraints. The lack of a healthy control group without GJH makes it difficult to conclude the effects of hypermobility combined with pain on proprioception and muscle endurance. Only absolute errors were examined in the study; however, if variable and constant errors were also evaluated, the magnitude and direction of errors would have provided more useful information.

Conclusion

From the results of this study, it can be concluded that GJH individuals with NSNP have demonstrated increased cervical JPEs and reduced cervical muscle (flexor and extensor) endurance compared to asymptomatic. Beighton hypermobility scores showed a significant positive moderate correlation with cervical JPEs in the NSNP group and a significant moderate negative correlation with cervical muscle endurance in individuals with and without NSNP. In current clinical practice, therapists should evaluate cervical proprioception and muscle endurance, and these factors should be considered during the rehabilitation of neck pain patients with GJH.

Supplemental Information

Supplemental Information 1 Code book.

Click here for additional data file.

Supplemental Information 2 Study data.

Click here for additional data file.

Additional Information and Declarations

Competing Interests

Author Contributions

Human Ethics

Data Availability

The authors declare that they have no competing interests.

Ravi Shankar Reddy conceived and designed the experiments, performed the experiments, analyzed the data, prepared figures and/or tables, authored or reviewed drafts of the paper, and approved the final draft.

Jaya Shanker Tedla conceived and designed the experiments, performed the experiments, analyzed the data, prepared figures and/or tables, authored or reviewed drafts of the paper, and approved the final draft.

Mastour Saeed Alshahrani conceived and designed the experiments, performed the experiments, analyzed the data, authored or reviewed drafts of the paper, and approved the final draft.

Faisal Asiri conceived and designed the experiments, performed the experiments, analyzed the data, authored or reviewed drafts of the paper, and approved the final draft.

Venkata Nagaraj Kakaraparthi conceived and designed the experiments, performed the experiments, analyzed the data, authored or reviewed drafts of the paper, and approved the final draft.

The following information was supplied relating to ethical approvals (i.e., approving body and any reference numbers):

King Khalid University Ethics Committee reviewed and approved this study (ECM#2021-4404).

The following information was supplied regarding data availability:

The raw measurements are available in the Supplemental File.

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
