# Peer review of "Comparison and correlation of cervical proprioception and muscle endurance in general joint hypermobility participants with and without non-specific neck pain—a cross-sectional study"

_PeerJ, doi:10.7717/peerj.13097_

## Round 0.1 · original submission · Major Revisions

The findings in this study on cervical proprioception and muscle endurance in general joint hipermobility patients with and without neck pain are novel and interesting. However, the statistical approach is not well addressed (see comments of reviewer 1), possible effects were not considered (e.g. gender), and there is a lack of contrast/comparison between the main results of the research with previous studies, as well as possible explanations for these results. For more details, please refer to the reviewers' comments.

Reviewer 1 ·

Basic reporting

This is an interesting topic worthy of further investigation. However, the statistical approach it’s not well aborded. For the first aim, the authors compared the cervical joint position error and muscle endurance in two groups of people with general hypermobility with and without neck pain. For the second aim, the general joint hypermobility was correlated with the principal outcomes (i.e., cervical joint position error and muscle endurance. The correlation analysis applied separately for each group may not respond to the second aim. ). The correct way is using a General Linear Model (GLM). Using the GLM, the authors can check the interaction between group x Beighton scores for each outcome (i.e proprioception, and endurance).
Furthermore, the authors need to check the gender effect in the model (e.g., group x Beighton x gender). To apply the GLM, the author needs to check first the potential assumption violations. Please consider these recommendations and recheck your results.

Experimental design

Line 98. The exclusion criteria need to be expanded. What about the neurological disease and Ehlers-Danlos Syndrome?
Line 112. Beighton score. It’s unclear whether the authors analyzed the dominant or both sides. Please clarify.
Line 114 and 133. Does the same operator make all assessments? Was the operator blinded to the group condition? Please clarify.
Line 135. Would you please give the error of the digital inclinometer used related to a gold standard
(e.g., 3D kinematic analysis).
Line 173. It`s unclear how the post-hoc analysis was made. They are only two groups (with and without neck pain), therefore only one comparison. Did the authors correct the p-value for multi-correlation analysis?

Validity of the findings

Results:
Please consider including figures to show the individual data and a 95 % confidence interval to illustrate the differences observed between groups.
Limitations:
The limitation needs to be expanded. The ideal would have been to have three groups. The lack of a healthy control group makes it difficult to conclude the effects of hypermobility combined with pain on proprioception and muscle endurance.
The association is not causation. Therefore, the authors need to have cautious about the interpretation and extrapolation of the results.

Additional comments

Abstract
Line 23. It does not clear the link between the first and the second sentences. The authors need to give the relevance to study the cervical joint position error and muscle endurance in people with general hypermobility with and without neck pain.
Introduction
Line 63-69. Please consider giving references to each sentence.
Line 68. Please consider using “therefore or consequently” instead of “so.”
Line 153. 2kg need to be separated (2 kg)

Reviewer 2 ·

Basic reporting

Clear, with some minor noun-verb agreement and definite/indefinite article problems.
These are listed below by their line number.

Experimental design

Solid and appropriate

Validity of the findings

Findings here are novel and of scientific importance, but a more developed attempt at an explanatory account is needed. This account does not have to be right, just consistent with the current and previous findings, and generative of future research.

Additional comments

The findings in this study on cervical proprioception and muscle endurance in GJH patients with and without neck pain are novel and interesting, combining significant differences between groups and correlational findings regarding the Beighton score. However, the findings have scientific significance beyond the clinical significance that is emphasised in the conclusion, and deserve a more concerted explanatory effort from the researchers here. This additional explanation should also involve consideration of the findings on cervical endurance, proprioception, range of motion and subclinical neck pain reported in the papers by Lee et al [Lee, H., Nicholson, L. L., & Adams, R. D. (2004). Cervical range of motion associations with subclinical neck pain. Spine, 29(1), 33-40. Lee, H., Nicholson, L. L., & Adams, R. D. (2005). Neck muscle endurance, self-report, and range of motion data from subjects with treated and untreated neck pain. Journal of manipulative and physiological therapeutics, 28(1), 25-32. Lee, H., Nicholson, L. L., Adams, R. D., & Bae, S. S. (2005). Proprioception and rotation range sensitization associated with subclinical neck pain. Spine, 30(3), E60-E67.] While this work did not involve GJH participants, it involves similar findings with respect to exactly the same test of cervical endurance, but different findings regarding proprioception, where pain was associated with greater proprioceptive sensitivity.
By including a review of this work, plus some account of the findings here with GJH participants, a more integrated view of the cervical system under stress from hypermobility and pain could be developed.


Specific edits needed:
l.26 to assess (rather than ‘see’)
l.42 Also, a moderate negative correlation …….was found. (currently, there is no verb in the sentence)
l.46 More explanation of this finding is needed
l.57 with women being five times
l.61 more mobility than stability
l.66-68 Previous research? Please cite some at the end of the sentence
l.78 In the literature, researchers have shown that
l.80 These findings suggest that
l.85 2) to assess the strength of the relationship
l.97 Participants were included
l.101 physical therapist’s directions.
l.109 has been used globally
l.112 was the most common
l.118 had to reposition to.
l.127 reaches what he or she thinks
l.130 A single examiner
l.159 2) omit ‘when’, as there is an ‘if’ before 1)
l.296 Pearson’s correlation analysis
l.217 Table 3
l.221 and to correlate
l.289 may be
l. 304-306 This finding is novel, and deserves more explanation

---

## Round 0.2 · Major Revisions

Although the authors of the manuscript addressed a large number of comments and suggestions from reviewers, key issues still need to be addressed. Two points are the most relevant:

i) the statistical analysis, although consistent with the general objective, is insufficient for the factors (e.g. group and gender) investigated, discussion, and especially the conclusion drawn. Please address this issue following the comments of Reviewer 1;

ii) some state-of-the-art aspects of the problem were not used in the current version of the manuscript (reviewer 2). For these reasons, we invite the authors to address these comments again and provide a solution through a new review.

Reviewer 1 ·

Basic reporting

Unfortunately, the authors do not respond to all requirements.
My major issue is about the statistical approach to respond if there is a relationship between GJH Beighton score and cervical JPS scores, neck muscle endurance capabilities in participants with and without NSNP. The authors need to check the interaction between conditions, as well as the interaction between gender. Again, my suggestion is to check if the conclusion of the study is still the same if the author makes a statistical analysis using another approach as General Linear Model. The authors should verify if there is an interaction between group x Beighton scores for each outcome (proprioception and endurance). Furthermore, the authors need to verify if gender influences the result in an interaction between group x Beighton x gender. Finally, the authors need to be sure that there is no statistical error type I in their conclusions.

Minor comments:
Same comment by the first revision. “It`s unclear how the post-hoc analysis was made. They are only two groups (with and without neck pain), therefore only one comparison. Did the authors correct the p-value for multi-correlation analysis?”. Please verify the accuracy of the sentence.
“Post-hoc analysis using the Bonferroni analysis was done for the variables which showed a statistically significant difference”

Experimental design

No comment.

Validity of the findings

No comment.

Additional comments

No comment.

Reviewer 2 ·

Basic reporting

The English definite article is used inappropriately on several occasions. I have detailed these.

Experimental design

No comment

Validity of the findings

No comment

Additional comments

In my original review, I asked the authors to look at some work by Haejung Lee and colleagues, on neck extensor endurance and cervical proprioception, to integrate into the current manuscript.
No mention was made of this work, so I thought that the authors could not find it, until I found a paper by Reddy et al (Journal of Bodywork and Movement Therapies, 2021), where, in their section 2.4.2, they stated that the endurance testing protocol that they followed came from Lee et al (2005). So, in the present paper in Line 205 I would ask that the authors acknowledge that the actual source of the testing protocol was Lee et al (2005), as is done in Reddy et al (2021).
Reddy, R. S., Meziat-Filho, N., Ferreira, A. S., Tedla, J. S., Kandakurti, P. K., & Kakaraparthi, V. N. (2021). Comparison of neck extensor muscle endurance and cervical proprioception between asymptomatic individuals and patients with chronic neck pain. Journal of Bodywork and Movement Therapies, 26, 180-186.


In addition, the following specific edits are needed.
Specific Edits:
L.67 No English definite article needed
L.111 to evaluate the relationship
L.150 has been utilized
L.178 No English definite article needed
L.179 No English definite article needed
L.205 test was adapted from Lee et al (2005)
L.265 No English definite article needed
L.279 correlation [not connection]
L.298 no significant correlations
L.311 No English definite article needed
L.325 reflexively
L.326 omit ‘vicious’
L.382 From the results of this study

---

## Round 0.3 · Minor Revisions

The authors of the manuscript addressed the comments and suggestions from reviewers. Before the article is accepted, it is necessary that on line 38 of the revised text please use the plural noun 'times' and not 'timings''.

Reviewer 1 ·

Basic reporting

The authors have made all suggested changes.

Experimental design

No comment

Validity of the findings

No comment

Additional comments

No comment

Reviewer 2 ·

Basic reporting

No comment

Experimental design

No comment

Validity of the findings

No comment

Additional comments

In line 38 of the revised text, please use the plural noun 'times' and not 'timings''

---

## Round 0.4 · accepted · Accept

Comparison and correlation of cervical proprioception and muscle endurance in general joint hypermobility participants with and without non-specific neck pain – a cross-sectional study - has been Accepted for publication. Congratulations!